# Incorporating Uncertainty Quantification for the Performance Improvement of Academic Recommenders

**Jie Zhu** 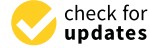**, Luis Leon Novelo and Ashraf Yaseen** *

Department of Biostatistics and Data Science, School of Public Health, The University of Texas Health Science Center at Houston, Houston, TX 77030, USA
* Correspondence: ashraf.yaseen@uth.tmc.edu

**Abstract:** Deep learning is widely used in many real-life applications. Despite their remarkable performance accuracies, deep learning networks are often poorly calibrated, which could be harmful in risk-sensitive scenarios. Uncertainty quantification offers a way to evaluate the reliability and trustworthiness of deep-learning-based model predictions. In this work, we introduced uncertainty quantification to our virtual research assistant recommender platform through both Monte Carlo dropout ensemble techniques. We also proposed a new formula to incorporate the uncertainty estimates into our recommendation models. The experiments were carried out on two different components of the recommender platform (i.e., a BERT-based grant recommender and a temporal graph network (TGN)-based collaborator recommender) using real-life datasets. The recommendation results were compared in terms of both recommender metrics (AUC, AP, etc.) and the calibration/reliability metric (ECE). With uncertainty quantification, we were able to better understand the behavior of our regular recommender outputs; while our BERT-based grant recommender tends to be overconfident with its outputs, our TGN-based collaborator recommender tends to be underconfident in producing matching probabilities. Initial case studies also showed that our proposed model with uncertainty quantification adjustment from ensemble gave the best-calibrated results together with the desirable recommender performance.

**Keywords:** uncertainty quantification; Monte Carlo dropout; ensemble; recommender systems; BERT; temporal graph networks



## 1. Introduction

With deep learning (DL) being widely applied in many areas of decision-making, such as investment opportunities, medical diagnosis, and recommendations, it is therefore of critical importance to evaluate the efficacy of these methods before their real-world application [1], especially when the cost/risk is high.

Uncertainty quantification (UQ) is a measure for evaluating the reliability and trustworthiness of model predictions. Uncertainty usually depends on the quantity, quality, and relevance of data and on the relevance and reliability of models and inferences used to fill the gaps [2]. In other words, uncertainty comes from two sources: data (aleatoric) uncertainty and model (epistemic) uncertainty [3–6]. While the former is an inherent property of data distribution and therefore irreducible, the latter occurs due to inadequate knowledge and is hence reducible.

Despite the notable accuracies of DL models in supervised learning benchmarks, DL tends to produce poorly calibrated results, which are either overconfident or underconfident [3,4,7]. Poorly calibrated results can be harmful in real-world applications; thus, it is essential to monitor UQ in a proper manner so that uncertain results can be either ignored or passed onto human experts for further handling. Unfortunately, characterizing uncertainty over parameters of deep learning networks is challenging due to the high dimensionality of the weight space and, potentially, the complex dependencies among

them [8]. Efforts have been made in the UQ of deep learning models: some through (the tractable approximation) of Bayesian neural networks and inferences [5,9–14]; among them, Monte Carlo (MC) dropout [10] is one of the most popular approximations which works by considering all possible outcomes from the distribution of decision boundaries. On the other hand, ensemble methods [15–17] are another commonly used technique by constructing multiple deterministic neural networks and averaging the results.

Most of these UQ studies have focused on computer vision and image processing tasks using benchmark datasets such as ImageNet, MNIST, etc., instead of real-life applications. Kennamer et al. [18] empirically studied MC dropout in an astronomical observing condition using simulated images and found that UQ resulted in improved accuracy and better calibrated results. Ng et al. [19] evaluated Bayes by backdrop, MC dropout, deep ensembles, and stochastic segmentation networks for MRI segmentation tasks in terms of accuracy, calibration, uncertainty on out-of-distribution datasets, and quality control on two benchmark datasets. Lakshminarayanan et al. [15] demonstrated that deep learning ensembles produced reliable uncertainty estimates via both regression and classification experiments on ImageNet. Ovadia et al. [20] presented a large-scale benchmark comparison of different UQ estimates under dataset shift. Additionally, we also found one study [21] that utilized the UQ technique on a deep learning model and further proposed methods to reduce uncertainties for the failure rate prediction of water distribution networks. However, there are few studies that have centered around natural language processing (NLP), and there are even fewer studies in recommender systems; the only studies that we found were in [22–25]. Zeldes et al. [22] proposed a mixture density network model to estimate the uncertainty of their online recommender platform. Shelmanov et al. [23] compared various MC dropout approaches to quantify uncertainties of a natural language understanding model on the General Language Understanding Evaluation benchmark dataset [26]. Penha and Huaff [24] proposed BERT-based stochastic rankers and showed that uncertainty estimation was beneficial for both risk-aware neural ranking and for predicting unanswerable conversational contexts. Siddhant and Lipton [25] empirically showed that the uncertainty estimates provided by MC dropout or Bayes by backdrop proved effective for active learning tasks on several standardized NLP datasets. Furthermore, it is unclear how they would utilize this UQ information to aid or improve current decision-making processes.

In this study, we built on previous experiments of our virtual research assistant (VRA) platform, a web-based recommender with different deep-learning components, and introduced methods to quantify the uncertainty associated with our model's predictions. We aimed to understand our model's behavior and use UQ to improve the results for our DL-based recommendation system. Specifically, our main contributions are as below:

- First, we carried out UQ experiments on different deep learning components of our virtual research assistant (VRA) platform, a web-based recommender for population health professionals that recommends datasets, grants, and collaborators. (The service platform is available at http://genestudy.org/recommends/#/ (accessed on 6 April 2023). We utilized two widely accepted UQ methods: MC dropout and ensemble. Using UQ, we were able to better understand the behaviors of our recommender outputs, while our grant recommender (BERT-based) tends to be overconfident with its outputs, and our collaborator recommender (TGN-based) tends to be underconfident in producing matching probabilities.
- Secondly, we introduced a new metric to incorporate the UQ information into our ranking scores. With this information, we were able to down-rank recommendations that the models were less sure about, thus reducing the risks associated with uncertain recommendations for better user experience. Moreover, the evaluations revealed that our proposed method with ensemble was able to produce consistently better results in a variety of metrics including calibration.

The rest of the article is organized as follows: an overview of the data used in the experiments is summarized in Section 2. It includes the collected grant and publications for the grant recommender, as well as the publication dataset for the collaborator recommender.

Section 3 shows the methods used for developing the different components of the VRA, uncertainty quantification, utilization of UQ information in the recommendation ranking, and evaluation, and they are described in the Methods section. Section 4 reports the experimental results and the detailed analysis. Finally, the conclusions, discussion, and future directions are discussed in Section 5.

## 2. Data

Depending on different recommendation tasks, for the grants and datasets (since the grants and datasets followed the same structures, we only presented the results of the grants recommender for conciseness of the content) vs. collaborators, we utilized different sets of data, similar to the previous experiments detailed in [27–30].

For the grants recommender, both grant metadata and the users' profiles built on publications for which grants will be recommended were needed. Metadata of grants (or research funding announcements (RFAs)), i.e., titles and descriptions, were collected from the NIH grants website (https://grants.nih.gov/funding/searchguide/index.html#/ (accessed on 6 April 2023)) (Figure 1), and researcher publications, i.e., titles and abstracts, were collected from the MEDLINE databases using PubMed (Figure 2).

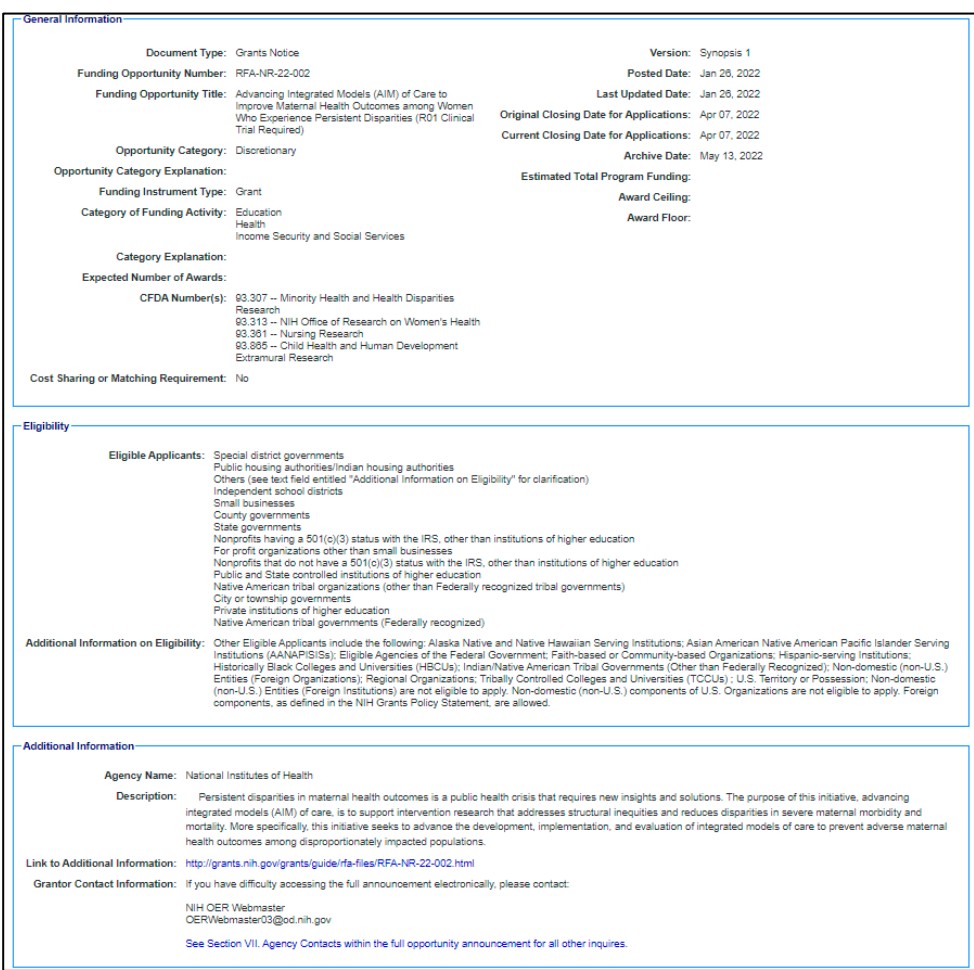

**Figure 1.** An example of NIH grant details.

The recommendation was then formulated as a classification task where the true pairs were created based on existing grant citations within the publications. More specifically, the relationships between the MEDLINE articles and the RFAs were established via the NIH's ExPORTER (https://exporter.nih.gov/ (accessed on 6 April 2023)). It archives relationships between publications and project numbers of funded grants, as well as relationships

between project numbers and corresponding RFAs. Using these two relationships, we established the relationships between the publications and NIH grants for evaluation. This relationship was then processed into a citation dictionary with each entry recorded as {'1287764': [PAR-17-095, PAR-12-298]}, where '1287764' is the PubMed Identifier (PMID) and 'PAR-17-095' and 'PAR-12-298' are the two RFAs that are associated with this publication. False pairs were then created using negative combinations. (For further details, see our previous paper [28]). We used random splits of 7:1:2 for training, validation, and testing (Table 1).

> J Voice. 2018 Nov;32(6):660-667. doi: 10.1016/j.jvoice.2017.09.014. Epub 2017 Oct 31.

## The Vocal Tract Organ: A New Musical Instrument Using 3-D Printed Vocal Tracts

David M Howard [1]

Affiliations  − collapse

**Affiliation**

1  Department of Electronic Engineering, University of London, Surrey, United Kingdom. Electronic address: david.howard@rhul.ac.uk.

**Abstract**

The advent and now increasingly widespread availability of 3-D printers is transforming our understanding of the natural world by enabling observations to be made in a tangible manner. This paper describes the use of 3-D printed models of the vocal tract for different vowels that are used to create an acoustic output when stimulated with an appropriate sound source in a new musical instrument: the Vocal Tract Organ. The shape of each printed vocal tract is recovered from magnetic resonance imaging. It sits atop a loudspeaker to which is provided an acoustic L-F model larynx input signal that is controlled by the notes played on a musical instrument digital interface device such as a keyboard. The larynx input is subject to vibrato with extent and frequency adjustable as desired within the ranges usually found for human singing. Polyphonic inputs for choral singing textures can be applied via a single loudspeaker and vocal tract, invoking the approximation of linearity in the voice production system, thereby making multiple vowel stops a possibility while keeping the complexity of the instrument in reasonable check. The Vocal Tract Organ offers a much more human and natural sounding result than the traditional Vox Humana stops found in larger pipe organs, offering the possibility of enhancing pipe organs of the future as well as becoming the basis for a "multi-vowel" chamber organ in its own right.

**Keywords:** 3-D printing; MRI; Vocal tract; Vowels; Vox humana.

**Figure 2.** An example of a MEDLINE article from PubMed.

**Table 1.** Grant recommender: training, validation, and testing data.

| Splits | # of Unique Publications | # of Records |
| --- | --- | --- |
| Training (7) | 135,766 | 216,766 |
| Validation (1) | 17,456 | 28,056 |
| Testing (2) | 40,730 | 65,104 |

For the collaborator recommender, MEDLINE articles were crawled from PubMed. In consideration of computational costs, we sampled only the years of 2019–2020 for the UQ experiments. Collaborations were defined as 'two or more authors sharing a publication'. Thus, temporal links were created using pairs of authors from each paper in the crawled data in chronological order. We defined the nodes as authors, and links as collaborations, with the timestamp of the links explicitly represented using the publication date, and raw node features using term frequency inverse document frequency (TF-IDF) calculated from titles of publications (For further details, see our previous paper [29]). Based on feedbacks from external evaluations in the previous paper, we used improved data processing to limit the influences of papers with long author lists in constructing our temporal graph: we only took the first three authors and the last corresponding author for creating collaboration links. We used chronological splits of 7-1.5-1.5 for training, validation, and testing, the same practice as seen in [31,32]; negative sampling of links was created during training. The basic summary statistics can be seen in Table 2.

**Table 2.** Collaborator recommender: training, validation, and testing data.

| Splits | # of Links | # of Nodes |
|---|---|---|
| Training (7) | 9589 | 9358 |
| Validation (1.5) | 1754 | 1796 |
| Testing (1.5) | 1559 | 1574 |

## 3. Methods

All implementation details can be found at https://github.com/ashraf-yaseen/VRA/tree/master/uncertainty_rec/ (accessed on 6 April 2023).

### 3.1. Our Virtual Research Assistant (VRA) Architecture

The experiments were carried out on our current version of VRA [27–30] where it has different components for recommendations. Specifically, the underlying modelling component for the datasets and grant recommendations is a BERT-based model, and the one for collaborators is a temporal graph network (TGN), see Figure 3. The detailed model architecture can be found in [27–30].

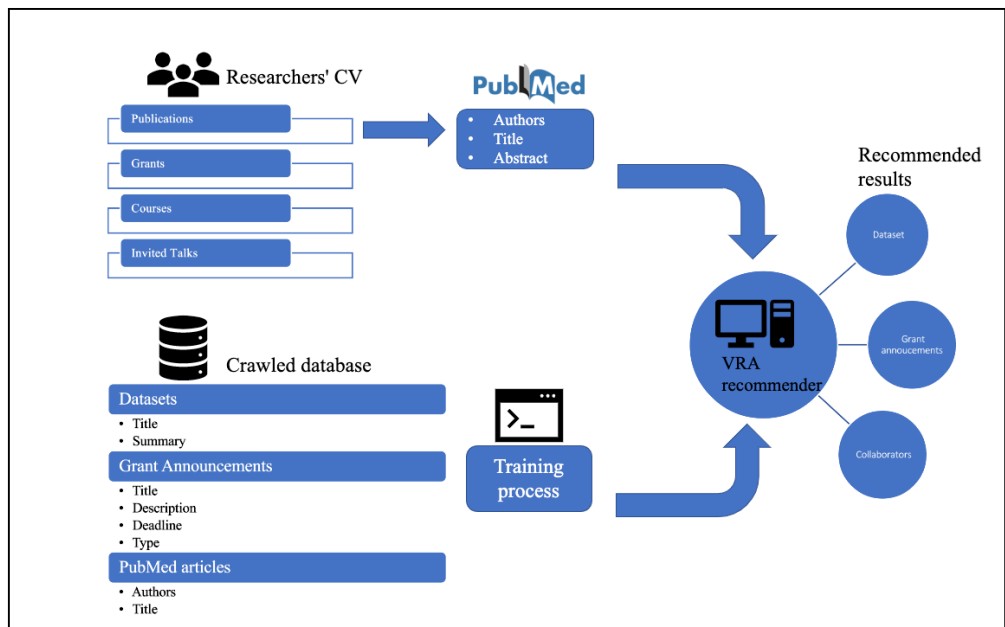

**Figure 3.** Our VRA architecture. The bottom part is the offline training stage, where the crawled database is used to train different deep learning components of our VRA: datasets, grants, and collaborators. The top part is the service stage, where the trained VRA takes in users' CV, feeds the information to the trained VRA, and provides recommendations based on the CV content.

#### 3.1.1. Grant Recommender: BERT-Based

We used sentence-pair classification formulation to calculate the probability of a PubMed article and a grant/RFA being a pair, see Figure 4. The output logits were then converted to probability and then used for aggregating and ranking the results for a particular article. For more details, see [28].

#### 3.1.2. Collaborator Recommender: TGN-Based

We represented authors as nodes, and collaborations (defined earlier in the Data section) as temporal links with publication years explicitly expressed as the time stamps of these links. Using TGN for producing node embeddings, we then calculated the edge probability at each time stamp for collaboration recommendation probabilities using a small neural network; see Figure 5. For more details, see [29].

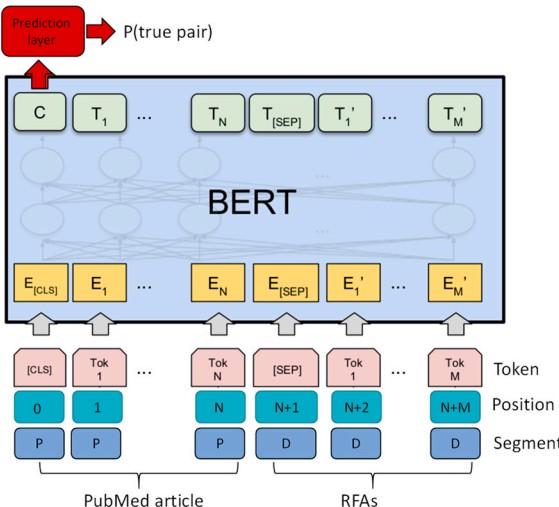

**Figure 4.** BERT architecture usage in our grant recommender.

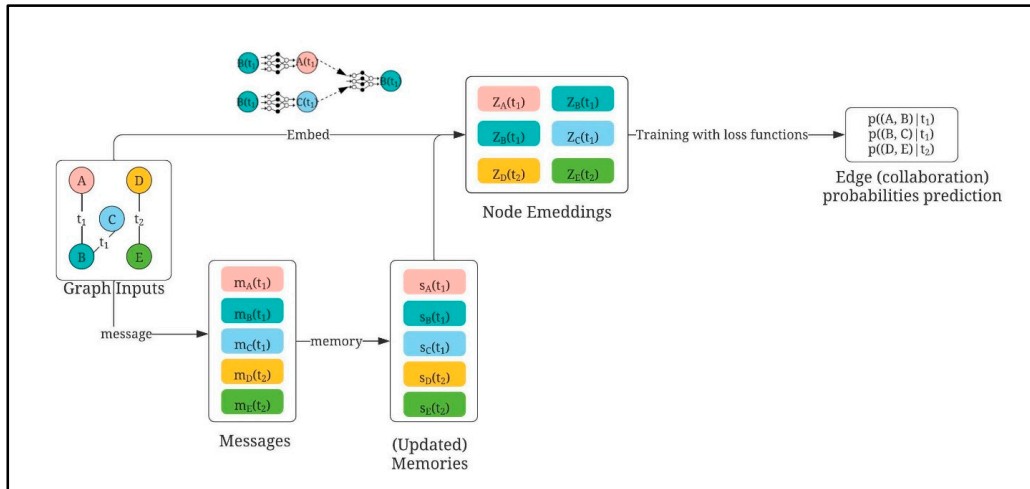

**Figure 5.** TGN architecture in our collaborator recommender, modified from [31]. Nodes A, B, C, D, and E are the authors, first constructed by using either (1) mesh terms or (2) the titles of the publications. Temporal links are defined by the 'sharing of publications' at time $t$. During training, each temporal collaboration is computed within a message between the involved nodes (e.g., at time $t_2$, the collaboration between D and E is calculated in both message $m_D(t_2)$ and in message $m_E(t_2)$). Then, the memory state of each author is updated using those temporal messages. The updated memories, together with embeddings constructed similar to GraphSAGE (concatenation of neighbor embeddings and its own embeddings) are then aggregated as the final embedding for each node. Then, the question of collaboration predictions becomes: given the embeddings of two nodes at time $t$, how likely will there be a link between them?

### 3.2. Uncertainty Quantification Methods

We experimented with both Monte Carlo (MC) dropout and ensemble methods for quantifying uncertainty.

### 3.2.1. Monte Carlo (MC) Dropout

Monte Carlo [33] sampling is an effective method for approximation when the exact posterior inference is intractable. In deep neural networks, dropout samples binary variables for each input data for every unit in the hidden layer with a probability to prevent overfitting [34]. It has been shown that enabling MC dropout during inference works approximately as Bayesian variational inferences for deep neural network models [10]. For

graph neural networks, however, dropout between the dense layers only is not enough, and therefore, we need to further introduce drop edges in the graph message passing in addition to regular dropout [11–13].

For each recommender component (grant vs. collaborator), we trained the model with the parameter $\theta$ and made $T = 100$ forward passes during the test time while keeping the dropout probability $p = 0.1$ for the BERT-based recommender; the dropout and drop edge probabilities were both $p = 0.1$ for the GNN-based recommender. Thus, each of the $T$ forward passes produced a prediction result:

$$\hat{y}^t = f(\theta; X_{test})^t; t = 1, \ldots . T \tag{1}$$

Then, the mean of predicted values $\bar{y} = \frac{1}{T}\sum_{t=1}^{T}\hat{y}^t$ is used as the final prediction and sample standard deviation $s_y = \sqrt{\frac{\sum_{t=1}^{T}\left(\hat{y}^t - \bar{y}\right)^2}{T-1}}$ of the measure of uncertainty.

### 3.2.2. Ensemble

Lakshiminarayanan et al. [15] proposed using the ensemble of deep neural networks to achieve uncertainty quantification. The idea is similar to the bootstrapping aggregation that is commonly used in the traditional machine learning and statistical learning methods such as random forest [35]. The ensemble (multiple models) is usually built using different initialization states and permutation of training data, if applicable. It is widely used due to its ease of implementation. However, the computational costs involved with deep learning ensembles are obviously high. The authors in [15] suggested using random initialization for scalability and that $M = 5$ ensembles is often good enough in practice.

In our case, for each recommender component, we trained $M = 5$ models with different random seeds without changing the actual training, validation, and test data, and each model thus had a different set of parameters $\theta_m, m = 1, \ldots M$. We used each of the $M$ models to predict on the same data for the prediction:

$$\hat{y}_m = f(\theta_m; X_{test}); m = 1, \ldots M \tag{2}$$

Then, the mean of the predicted values $\bar{y} = \frac{1}{M}\sum_{m=1}^{M}\hat{y}_m$ is used as the final prediction and the sample standard deviation $s_y = \sqrt{\frac{\sum_{m=1}^{M}\left(\hat{y}_m - \bar{y}\right)^2}{M-1}}$ of the measure of uncertainty.

### 3.3. Proposed UQ Adjusted Results

For UQ, we have the sample standard deviation from the calculations either from MC dropout or ensemble. In the statistical domain, standard error is usually used in constructing confidence intervals to represent how confident we are about certain parameter estimates [36]. Based on this idea, we propose using the following formula to take into consideration the UQ information of the final ranking score of the recommended results:

$$\hat{y}_{uq} = \bar{y} * exp\left(-c * s_y\right) \tag{3}$$

Here, $\hat{y}_{uq}$: UQ adjusted results and $\bar{y}$: original mean of the predicted values (logits), either from MC dropout or ensemble. $c$ is some constant, in our case, we choose 3.92, a commonly used number for constructing confidence intervals for standard normal distribution (twice the 97.5 percentile of the standard normal distribution). $s_y$ indicates the standard rror.

With the sense that the bigger the standard deviation, the more unsure our results are, and therefore, the corresponding recommendation scores should be downranked.

### 3.4. Evaluation Metrics

We compiled a list of commonly used metrics in recommenders for evaluation of the results for both grant and collaborator recommendation. We first supplemented the confusion matrix as Table 3 to better explain some metrics.

**Table 3.** Confusion matrix for the recommenders.

|  | Recommended | Not Recommended |
| --- | --- | --- |
| **Relevant** | True positives (TP) | False negatives (FN) |
| **Not relevant** | False positives (FP) | True negatives (TN) |

- AUC: A receiver operating characteristic curve, or ROC curve, is a graphical plot that illustrates the diagnostic ability of a binary classifier system as its discrimination threshold is varied. The AUC is the area under a ROC curve, which provides an aggregated measure of performance across all possible classification thresholds [37].
- Average precision (AP): This summarizes precision-recall curve as the weighted mean of precisions achieved at each threshold, with the increase in recall from the previous threshold used as the weight.
- Mean reciprocal rank (*MRR*): The reciprocal rank (RR) measures the reciprocal of the rank at which the first relevant document was retrieved. RR is 1 if the relevant document was retrieved at rank 1, RR is 0.5 if the document is retrieved at rank 2, and so on. When we average the retrieved items across the queries $Q$, the measure is called the *MRR*.

$$MRR = \frac{1}{|Q|} \sum_{i=1}^{|Q|} \frac{1}{rank_i} \qquad (4)$$

- Recall@1 (R@1): At the $k$-th retrieved item, this metric measures the proportion of relevant items that are retrieved. We evaluated recall@1.

$$Recall@k = \frac{TP@k}{TP@k + FN} \qquad (5)$$

- Precision@1 (P@1): At the $k$-th retrieved item, this metric measures the proportion of the retrieved items that are relevant. In our case, we are interested in precision@1.

$$Precision@k = \frac{TP@k}{TP@k + FP@k} \qquad (6)$$

- Expected calibration error (ECE): Calibration measures the discrepancy between long-run frequencies and subjective forecasts [38,39]. To make use of uncertainty quantification methods, we need to make sure the (binary) classifier estimates are as close to perfect calibration as possible, meaning that if we discretize our model predictions in $L$ interval bins, then we expect that the fraction of positives and predicted probabilities of each bin should agree. Mathematically, let $B_l$ be the set of samples whose predicted probabilities fall into interval $I_l = \left( \frac{l-1}{L}, \frac{l}{L} \right]$; the fraction of positives for $B_l$ is:

$$pos(B_l) = \frac{1}{|B_l|} \sum_{i \in B_l} y_i \qquad (7)$$

where $y_i$ is the true class label for sample $i$. The predicted probability within bin $B_l$ is:

$$predp(B_l) = \frac{1}{|B_l|} \sum_{i \in B_l} \hat{p}_i \qquad (8)$$

where $\hat{p}_i$ is the predicted probability for sample $i$.

*ECE* [40] is then one commonly used summary statistic that measures the difference between the expected probability and the fraction of positives:

$$ECE = \frac{1}{n}\sum_{l=1}^{L}\left|B_l\right|\sqrt{(pos(B_l) - predp(B_l))^2} \tag{9}$$

where *n* is the total number of samples.

In addition to *ECE*, we also plotted out calibration curves [41] as a visual tool to assess the agreement of $pos(B_l)$ with $predp(B_l)$ for each bin.

For the grant recommender, we calculated all six metrics. For the collaborator, we only calculated three metrics: AUC, AP, and ECE, since we analyzed the results in a temporal fashion, and therefore, aggregated results, such as MRR, etc., were not applicable.

## 4. Results

### 4.1. Grant Recommender: BERT-Based Results

We present the UQ results of the grant recommender below in Table 4. 'Regular' is the standard, deterministic model that we used in our previous experiments. 'MC dropout' and 'ensembles' are stochastic models with UQ applied, which output predictive distributions instead of point estimates. And finally, 'MC dropout, UQ adjusted', and 'Ensemble, UQ adjusted' are our proposed models with recommendation rankings further adjusted using UQ which are calculated through two corresponding methods.

**Table 4.** Grant recommender: BERT-based results for the regular model vs. the stochastic UQ models vs. the UQ adjusted proposed models.

| Models | AUC | AP | MRR | R@1 | P@1 | ECE |
|---|---|---|---|---|---|---|
| Regular | 0.977 | 0.975 | 0.933 | 0.810 | 0.871 | 0.073 |
| MC dropout | 0.978 | 0.977 | **0.947** | **0.816** | **0.882** | 0.067 |
| MC dropout, UQ adjusted | **0.979** | **0.978** | 0.939 | 0.816 | 0.882 | 0.064 |
| Ensemble | **0.981** | **0.980** | 0.941 | 0.818 | 0.884 | 0.067 |
| Ensemble, UQ adjusted | 0.975 | 0.967 | 0.938 | 0.815 | 0.879 | **0.030** |

For our grant recommender: BERT-based, we could see that in Table 4, almost all performance metrics for our stochastic models, either through MC dropout (2nd row) or ensemble (4th row), are better than the regular results. The calibration curve in Figure 6a. below further confirms the improved calibration by showing that the curves of both stochastic models are closer to the perfect calibration line than the regular models.

Our proposed method with UQ calculated from MC dropout yielded equally good, if not better results in terms of AUC, AP, R@1, and P@1 and lower ECE, as also confirmed in Table 4.

Our proposed method with the UQ calculated from ensemble yielded slightly worse results in AUC, AP, MRR, R@1, and P@1; the differences, however, were not substantial. Given that we took UQ into the final ranking decisions, we thus possibly changed the ordering in metrics that consider only probabilities. On the other hand, we were able to produce well calibrated results and reduced the ECE by more than 50%.

The regular BERT-based recommender in general produced overly confident predictions, as indicated by much higher mean predicted probabilities at both ends marked by the dark green curve in Figure 6a. as well as in the histogram in the same color as on the top row in Figure 6b, an issue that frequently happens in deep learning models [3,4,7]. But our proposed method with ensemble was able to produce the best calibrated results as shown by the light green curve (Figure 6a) and the more evenly distributed light green histogram on the bottom row (Figure 6b) and was able to use that uncertainty to achieve good recommender performance as well.

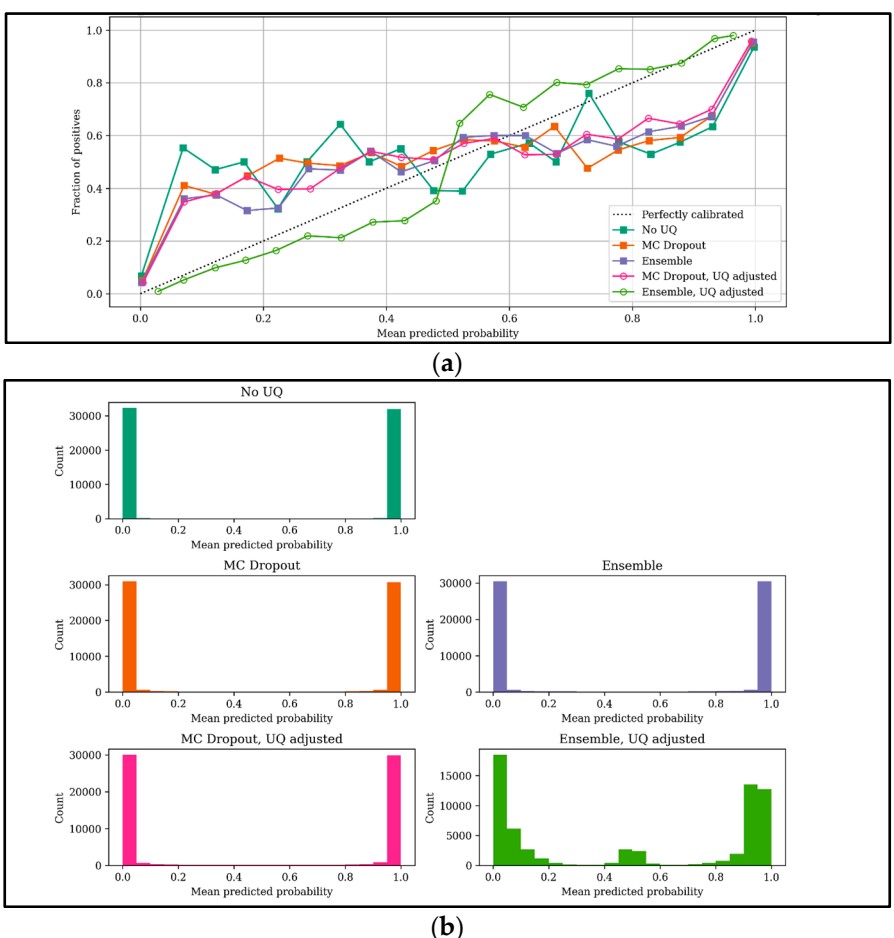

**Figure 6.** (**a**) BERT-based grant recommender, calibration plots for the regular model vs. the stochastic UQ models vs. the UQ adjusted proposed models. (**b**) BERT-based grant recommender, mean predicted probability count plots for the regular model vs. the stochastic UQ models vs. the UQ adjusted proposed models.

### 4.2. Collaborator Recommender: TGN-Based Results

We present the results of the grant recommender below in Table 5. For our TGN-based collaborator recommender, we could see that the behavior is a bit different from the BERT-based grant recommender. The performance metrics, such as the AUC and AP for the stochastic models, either through MC dropout (2nd row) or ensemble (4th row), are better than the regular results, at the expanse of calibration.

**Table 5.** Collaborator recommender: TGN-based results for the regular model vs. the stochastic UQ models vs. the UQ adjusted proposed models.

| Models | AUC | AP | ECE |
|---|---|---|---|
| Regular | 0.792 | 0.727 | 0.165 |
| MC dropout | 0.796 | 0.711 | 0.184 |
| MC dropout, UQ adjusted | **0.817** | **0.744** | **0.162** |
| Ensemble | 0.938 | 0.960 | 0.282 |
| Ensemble, UQ adjusted | **0.960** | **0.972** | **0.154** |

Our proposed method with UQ calculated from MC dropout yielded better results in AUC and AP and lower ECE, as also confirmed in Table 5.

Our proposed method with UQ calculated from ensemble yielded much better results in all metrics with the lowest ECE among all models (reduced by 10%), as also shown in Table 5.

Unlike our BERT-based grant recommender, our regular, deterministic TGN-based collaborator recommender tended to produce under-confident results, as the mean predicted probabilities center around the middle instead of both extreme ends. The phenomenon is reflected in the dark green curve (Figure 7a) and the dark green histogram on the top row (Figure 7b), an issue commonly found in deep-learning-based models [3]. Again, our proposed method with ensemble was able to produce the best-calibrated results, as shown both in the light green curve (Figure 7a) and the more evenly distributed light green histogram on the bottom right (Figure 7b), while at the same time, it was able to produce the best recommender performance.

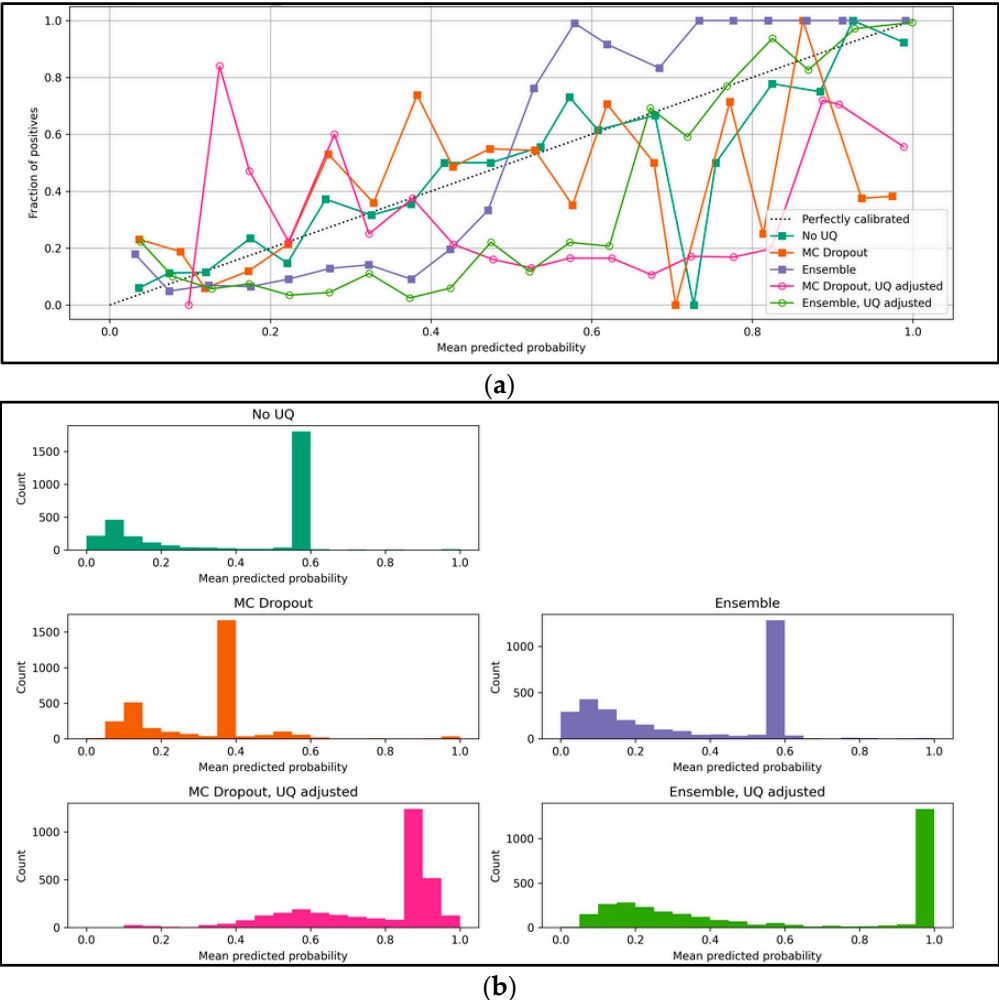

**Figure 7.** (**a**) TGN-based collaborator recommender, calibration plots for the regular model vs. the stochastic UQ models vs. the UQ adjusted proposed models. (**b**) TGN-based collaborator recommender, mean predicted probability count plots for the regular model vs. the stochastic UQ models vs. the UQ adjusted proposed models.

## 5. Conclusions and Discussion

In summary, we conducted UQ experiments on two separate components of our virtual research assistant (VRA) recommendation platform: the BERT-based grant recommender and the TGN-based collaborator recommender. For each component, we compared the recommendation predictions of (1) a regular, deterministic model, (2) stochastic UQ models using two techniques: MC dropout and ensemble, and (3) our proposed model that takes

in UQ calculated through two methods and adjusts the results accordingly, in terms of both recommendation metrics and calibration/reliability metrics. Using UQ, we were able to better understand the behavior of our regular recommender outputs: while our BERT-based grant recommender tends to be overconfident with its outputs, our TGN-based collaborator recommender tends to be underconfident in producing matching probabilities. The experimental results also showed that our proposed model with UQ calculated from ensemble was able to produce the most calibrated results also with the most desirable recommendation performance.

However, we do want to point out that instead of generalizing our results to other studies, our aim is to better understand our deep-learning-based recommender behaviors through UQ, a simple yet often ignored statistical concept, and to further use that information to improve the overall reliability of our recommender results for better user experience. More importantly, we hope to possibly provide a paradigm for those also working on real-life applications with real-life datasets. Nevertheless, there are courses of action that we want to work on further in our research group's continuing studies. One next step is to implement double blind evaluations on the platform and involve human judges/evaluators. We hope to acquire a better sense of whether our proposed adjustment also leads to better human ratings. Secondly, the choice of value $c$ in our proposed method could be systematically analyzed. We currently only experimented with a limited number of $c$ values empirically and found out that 3.92, which corresponds to twice the 97.5 percentile of the standard normal distribution, worked empirically well. However, it should be treated as a hyperparameter and tuned more rigorously in future studies.

**Author Contributions:** Conceptualization, J.Z., L.L.N. and A.Y.; methodology, J.Z., L.L.N. and A.Y.; software, J.Z.; validation, J.Z., L.L.N. and A.Y.; formal analysis, J.Z.; investigation, J.Z.; resources, A.Y.; data curation, J.Z.; writing—original draft preparation, J.Z.; writing—review and editing, J.Z., L.L.N. and A.Y.; visualization, J.Z.; supervision, A.Y.; project administration, J.Z.; All authors have read and agreed to the published version of the manuscript.

**Funding:** This research received no external funding.

**Institutional Review Board Statement:** Not applicable.

**Informed Consent Statement:** Not applicable.

**Data Availability Statement:** The Python code, the demo datasets, and the supplemental materials relevant to this article are publicly available at: https://github.com/ashraf-yaseen/VRA/tree/master/uncertainty_rec (accessed on 6 April 2023).

**Acknowledgments:** The authors would like to thank Hulin Wu, Jose-Miguel Yamal, Hongyu Miao, and George Delclos for the insightful discussions.

**Conflicts of Interest:** The authors declare no conflict of interest.

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
