# Peer review of "Incorporating Uncertainty Quantification for the Performance Improvement of Academic Recommenders"

_knowledge, doi:10.3390/knowledge3030020_

Round 1

Reviewer 1 Report (Previous Reviewer 1)

I was actually one of the previous reviewer of this manuscript submitted to another journal. I just want to provide my previous comments on this draft and many points of them I think still holds:

This paper provides uncertainty quantification method using dropout and ensemble to improve existing deep learning based recommendation systems.  A new metric is introduced to use the quantified uncertainty to improve the model performance based on their fidelity of the outputs. Some motivations of using uncertainty quantification to enhance the prediction and model performance are provided in the introduction part. High level motivations of the numerical experiments are given as well. The paper is clearly written but I think it is a bit lack of novelty here. This is because neither of these UQ methods are developed by the authors. The authors did not develop the model architecture neither. So, doing these two methods should be just simple change and the contribution could note be considered much novel. 

  1. Both ensemble and dropout are highly scalable methods for uncertainty quantifications in deep learning however, their performance yield only suboptimal in conducting uncertainty quantification. This is why, in particular for dropout, is usually used as a regularization tool instead of UQ tool. Given the fact that the systems the authors are considering is large and Hamiltonian MCMC could not be applied. I think the authors could still consider variational inference on the model’s parameter to compare with the proposed two methods. I think two methods presented in this manuscript is not very convincing as a systematic study. 
  2. The equation 3 is just a variant of lower confidence bound … which probably could not be considered as some novel contribution. Maybe the only difference is the exp() function. But I think this is not mathematically correct. Since both y and the s_y are having same unit. Taking an exp() function outside the uncertainty does not make good sense to me. 
  3. For some cases in table 4, the adjusted results are not better than vanilla version. Any intuition here?
  4. One importance aspect of recommendation system requires the system to be fast so that the latency does not affect the user’s experience a lot. In the cases that the authors consider, maybe latency is not very important. But providing a table to summarize training and prediction computational cost should be helpful for the readers to follow.
  5. Please increase the font size of the words in most images so that the readers can see the contents clearly. This is particularly useful in figure 1 and figure 2 which show how the training data are constructed.  

Author Response

Thanks so much for your comments and suggestions. Based on the review, we’ve made the following adjustments to further improve our manuscript. All changes can be found in Track Changes, All Markup option.  

  1.   Thank you so much for the comments.   

One thing to clarify about our work presented in this manuscript: we introduced uncertainty quantification through Monte Carlo dropout and ensemble to better understand the behaviors of two different components (i.e., BERT-based, Temporal Graph Network (TGN)-based) in our Virtual Research Assistant (VRA) recommender system, a platform for academic/scholarly recommendations (datasets, grants, collaborators etc.) to population health professionals. The goal is to help our VRA make better, more reliable recommendation results, considering the fact the deep learning-based architectures are usually either overconfident or under-confident in predictions. Our previously published papers could be found at [1]–[4]. But we do agree with the reviewer that more systematic comparison and analysis should be carried out, should generalization on uncertainty methods to deep learning-based recommenders be the desired goal. We are considering that as a future direction in our research.  

Based on published work here [5], Monte Carlo dropout is considered as mathematically equivalent to variational approximation of the Bayesian Neural Networks with Bernoulli Distribution that is more efficient than traditional variational inference. Hence the reason to choose MC dropout over variational approximation.  

  1. The reason we introduced exp() in eq(3) is that, we want the UQ to be a scale factor to our original logit predictions. In this case naturally, when the standard error is infinitesimal, let’s say lim->0, we want the prediction as it is, and exp (0) =1, which gives us what we were looking for. Otherwise, it will turn any ‘very sure’ yuq = 0, and apparently, we do not want that. We do admit that the equation is empirical for our purpose, and we acknowledge this fact in the last paragraph starting line 299.  

  1. For table 4, our adjusted results were better than the ‘regular’.  If by ‘vanilla’, the reviewer was referring to MC Dropout/Ensemble (vs the ones we adjusted accordingly), yes, we do expect some changes in classification/recommender metrics. The objective behind the experiments presented in this work is to make better, more reliable recommendation results, considering the fact that deep learning-based architectures are usually either overconfident or under-confident in predictions. The simplified scenario would be: if our VRA recommends grant A with possibility of match of 0.9 and uncertainty score of xa, and grant B with possibility of match of 0.85 and uncertainty score of xb (let’s say,  xa >>>>>> xb), so how should we rank these two recommendations and present them to users?  In our case, we try to take that UQ into account and not just look at the probability score alone (contrast to what other recommenders do). In doing so, we hope to provide more reliable recommendations, while at the same time, hopefully, do not decrease the actual performance metrics too much. And ECE is the primary metric to reflect that reliability. That is why, we were primarily focusing on ECE, while keeping check on other metrics such as AUC, AP that consider only probabilities. Even though they are slightly lower in Ensemble adjusted case, we do think it is within acceptable range considering the improvement in ECE. We also added this information in the paragraph starting 254.   

  1. Thanks so much for the suggestion. As the reviewer pointed out, our recommender was indeed fairly computationally expensive during training, but as our goal is the service for end-users, this process is usually super-efficient based on our previous experiments in [1]–[4]. In our previous papers, we did try to include all the necessary training/test time but were advised not to during revisions for publication since most reviewers saw them as a deviation from the main content and/or unnecessary. Thus, we did not record the time this time for experiments. We do apologize for this.  

  1. The font size of words in all images has been increased. In particular, the screenshots of NIH grant (Figure 1) and MEDLINE article (Figure 2) have been enlarged to show the meta content we used to construct our data. 

References:  

[1] J. Zhu, B. Patra, and A. Yaseen, “Recommender systems of scholarly papers using public datasets,” in 2021 AMIA Informatics Summit, Mar. 2021. 

[2] J. Zhu, H. Wu, and A. Yaseen, “Sensitivity Analysis of a BERT-based scholarly recommendation system,” in Proceedings of FLAIRS-35, May 2022, vol. 35. doi: https://doi.org/10.32473/flairs.v35i. 

[3] J. Zhu and A. Yaseen, “A Recommender for Research Collaborators Using Graph Neural Networks,” Front. Artif. Intell., vol. 5, 2022, Accessed: Oct. 02, 2022. [Online]. Available: https://www.frontiersin.org/articles/10.3389/frai.2022.881704 

[4] J. Zhu, B. G. Patra, H. Wu, and A. Yaseen, “A novel NIH research grant recommender using BERT,” PLoS One. 2023 Jan 17;18(1):e0278636. doi: 10.1371/journal.pone.0278636. PMID: 36649346; PMCID: PMC9844873.

[5] Y. Gal and Z. Ghahramani, “Dropout as a Bayesian Approximation: Representing Model Uncertainty in Deep Learning,” ArXiv150602142 Cs Stat, Oct. 2016, Accessed: Nov. 08, 2021. [Online]. Available: http://arxiv.org/abs/1506.02142 

Reviewer 2 Report (Previous Reviewer 3)

In this study, the authors studied an interesting topic of using uncertainty quantification to improve the performance of academic recommenders. Overall, the manuscript shows good organization and interesting results. However, to perfect the paper presentation, some revisions should be completed.

1.         Abstract should be improved. The authors emphasized a lot of background, which is not directly related to the authors’ works and contributions.

2.         The writing of this manuscript should be carefully checked and revised. Some references are not shown in the manuscript.

3.         The author should include recent progress in uncertainty quantification for deep learning frameworks. For example, https://doi.org/10.1016/j.ress.2023.109088 provides a good application for uncertainty quantification of infrastructure failure.

Author Response

Thank you so much for your comments. Based on the review, we’ve made the following adjustments to further improve our manuscript. All changes can be found in Track Changes, All Markup option. 

  1. The abstract has been revised with better organization and more content regarding our work and contributions.  
  2. Thank you for pointing that out. We went through all the references in the manuscript to make sure they are correctly cited.  
  3. We revised the introduction part to better summarize the recent progress in uncertainty quantification for deep learning frameworks, including the paper as suggested by the reviewer (in paragraph starting from line 59). Please note that at the time of our submission (Dec 2022), this paper was not yet available for references.   

Round 2

Reviewer 1 Report (Previous Reviewer 1)

I appreciate the authors response to the comments. I think the paper is a boarder line paper.

Author Response

Thanks so much for your comments.

This manuscript is a resubmission of an earlier submission. The following is a list of the peer review reports and author responses from that submission.

Round 1

Reviewer 1 Report

I appreciate the authors effort on revising the manuscript. I believe the authors tried their best. Now I think this paper is a boarder line paper and I do not have a strong opinion that it should be accepted or rejected. I would like to let the editor to make decision. 

Author Response

Thank you very much for your feedback. We revised the manuscript again, especially the method section as pointed out by the rating review. All changes can be tracked using 'all markup'.

Reviewer 2 Report

Thanks authors for their response and the corresponding changes. I appreciate the improved writing and additional materials.

After reading the response in detail, I believe my previous concerns still stands, i.e.,

1. Limited Novelty / Generality: the paper applies rather well-known uncertainty methods to a recommender problem with non-standard architecture (BERT/TGN) and a relatively small dataset. The proposed uncertainty adjustment scheme is rather obvious (Eq 3) and does not come with additional theoretical insight.

2. Narrow focus / lack of generality:  The generality of the result is a bit concerning. The paper is only illustrated on a specialized system on not very commonly used datasets. It is difficulty for me to imagine the result and conclusions can be straightforwardly generalized to other settings. As a result, I am not sure if the paper in its current form can benefit the intended  audience of the Informatics journal. To address this, please consider at least including a few additional, more commonly studied datasets into the study to illustrate the generality of the conclusions.

3. (Minor) re: "actionable insight". In order to not misleading readers, please consider not including claims in the title that the content of the paper cannot substantiate.  To this end, I see the authors mentioned in the response "which will **potentially** lead to actions or responses" without including additional experiment to substantiate such claim (e.g., evaluating if added uncertainty can be used to improve the  performance of downstream actions (e.g., abstention / selective prediction during recommendation)). Consequently, I would recommend not including such claims in the title.

Author Response

Thank you so much for your comments.

  1. We agree with the reviewer that our system is limited to the field of population health (hence the choice of the dataset in our work). But, the underling idea is generalizable to all fields. To the best of our knowledge, there are very few real-life recommenders that incorporate UQ. In the work presented in this manuscript, the goal is to improve the performance of our VRA in making better, more reliable recommendation results, considering the fact the deep learning-based architectures are usually either overconfident or under-confident in predictions. By utilizing statistics concepts, though simple, we were able to achieve better reliable results, which hasn’t been done before for real-life recommendation platforms.

In conclusion, we recommend that developers of recommendation systems employ UQ into their methods for more reliable results.

  1. When we first started working on our VRA platform, we picked datasets related to areas in population health because we are from a school of public health and our initial potential users are researchers/faculty in this area. This made it possible to get them involved in evaluating the system and provide feedback.

As for our work in the submitted manuscript, our goal as mentioned in the abstract is to improve the VRA recommendation results by considering UQ, and thus we did not intend our manuscript to be comprehensive and generalizable to all deep learning-based methods. However, we agree with the reviewer that more systematic comparison and analysis should be carried out, should generalization on uncertainty methods to deep learning-based recommenders be the desired goal. We are considering that as a future direction in our research. 

But we do think it is suitable for the intended audience due to the following:

  • Instead of ‘generalizations of our results’ to other deep-learning based recommenders, we hoped to provide a possible paradigm of practice for the real-life development and improvement of recommender systems, including the curation of practical datasets, as well as embracing the power of uncertainty quantification to enhance the recommendation results.
  • We don’t think it fits in our scenario to apply mainstream datasets (e.g. movielens), since they are hardly population health related practically, and they are rating-based and thus is intended for collaborative filtering, while ours is content based.Furthermore, much of the recommender systems were built on the theoretical base using ‘borrowed’ datasets. But studies[1]–[4]have shown that this much ‘heavy borrowing’ actually raises concerns about misalignment and overconfidence on the models’ ability to generalize on real-world scenarios.

  1. We considered the reviewer’s suggestion and felt it is appropriate to modify the title, and thus made the change as per reviewer’s comments.

New title: “Incorporating uncertainty quantification for performance improvement of academic recommenders”

References

[1]             B. Koch, E. Denton, A. Hanna, and J. G. Foster, “Reduced, Reused and Recycled: The Life of a Dataset in Machine Learning Research,” in Advances in Neural Information Processing Systems 35, Sydney, NSW, Australia, 2021, p. 13.

[2]             P. Lewis, P. Stenetorp, and S. Riedel, “Question and Answer Test-Train Overlap in Open-Domain Question Answering Datasets.” arXiv, Aug. 06, 2020. Accessed: Jun. 04, 2022. [Online]. Available: http://arxiv.org/abs/2008.02637

[3]             K. Blagec, G. Dorffner, M. Moradi, and M. Samwald, “A critical analysis of metrics used for measuring progress in artificial intelligence.” arXiv, Nov. 08, 2021. Accessed: Jun. 04, 2022. [Online]. Available: http://arxiv.org/abs/2008.02577

[4]             MIT news, “Nonsense can make sense to machine-learning models,” MIT News | Massachusetts Institute of Technology. https://news.mit.edu/2021/nonsense-can-make-sense-machine-learning-models-1215 (accessed Jan. 07, 2023).

Reviewer 3 Report

In this study, the authors studied an interesting topic of using uncertainty quantification to improve the performance of academic recommenders. Overall, the manuscript shows good organization and interesting results. However, to perfect the paper presentation, some revisions should be completed.

1.         Abstract should be improved. The authors emphasized a lot of background, which is not directly related to the authors’ works and contributions. 

2.         The writing of this manuscript should be carefully checked and revised. Some references are not shown in the manuscript.

3.         The author should include recent progress about the uncertainty quantification for deep learning frameworks. For example,https://doi.org/10.1016/j.ress.2023.109088 provides a good application for uncertainty quantification of infrastructure failure. 

Author Response

Thank you so much for your comments.

  1. The abstract has been revised with better organization and more content regarding our work and contributions.
  2. Thank you for pointing that out. We went through all the references in the manuscript to make sure they are correctly cited.
  3. We revised the introduction part to better summarize the recent progress in uncertainty quantification for deep learning frameworks, including the paper as suggested by the reviewer (in paragraph starting from line 59). Please note that at the time of our submission (Dec 2022), this paper was not yet available for references.

Reviewer 4 Report

The authors present a metric to incorporate uncertainty quantification in the recommendations generated by their algorithm. The paper is interesting but the throughout the paper there was an error message that made it difficult to follow the arguments.

The authors can improve the paper by first clarifying how they are presenting uncertainty on the user interface. The authors need to improve their discussion section by clarifying how they have improved the state of the art and what are the limitations of their findings. 

Author Response

Thank you very much for your comments.

We would like to clarify that uncertainty quantifications are calculated at the backend as an integral part of ranking process in the recommender before the recommended results are shown for users, so it’s not shown on the user interface.

We further modified/edited the abstract, overview as well as discussion and results sections on 1) our main contributions 2) our limitations, as can be seen throughout the content of revised manuscript.

Round 2

Reviewer 1 Report

I appreciate the authors efforts on revising the attached manuscript. I think the manuscript now is boarder line. I tend to reject since the math looks a bit ad-hoc to me. But I would leave the decision to the editor and the other review suggestions.

Reviewer 2 Report

Thanks authors for improving the paper. I appreciate your efforts.

As I have stated and confirmed my author, the central point of contention at this point is:

Whether the study on a specialized recommender system (whose data and model setting is not necessarily consistent with mainstream practice) is sufficient to guanrantee the broad generality of the claim (i.e., "developers of recommendation systems employ UQ into their methods for more reliable results.") to be informative for readership of Informatics. 

While I do agree with the message, as a potential audience, I do not feel the content and experiment in this paper is sufficient to convince me to consider UQ technique (outside model & data setting considered in this paper). As a  reseacher, I also do not find sufficient novelty on the uncertainty techniques & evaluation being considered. From this standpoint, I find this work may be more approperiate for a journal or workshop with a more specialized focus (e.g., Public Health Informatics). Consequently, I find it difficult to recommend acceptance of this work to a general-audience informatics journal.